Research on building extraction from remote sensing imagery using efficient lightweight residual network

Gao Ai
Yang Guang ygcmd@163.com
School of Information Engineering, Institute of Disaster Prevention , Sanhe , China
Alatas Bilal
Electronic publication date: 2024 May 2
Publication date: 2024
Volume: 10
Electronic Location ID: e2006
Received 2023 Nov 1; Accepted 2024 Apr 1
Copyright: ©2024 Gao and Yang
Copyright year: 2024
Copyright holder: Gao and Yang
License: This is an open access article distributed under the terms of the Creative Commons Attribution License, which permits unrestricted use, distribution, reproduction and adaptation in any medium and for any purpose provided that it is properly attributed. For attribution, the original author(s), title, publication source (PeerJ Computer Science) and either DOI or URL of the article must be cited.
License URL: https://creativecommons.org/licenses/by/4.0/

Keywords: ELRNet, Building extraction, Lightweight neural networks, Lightweight feature extraction modules, Very high-resolution remote sensing images

Funding: The National Natural Science Foundation of China 42007422 The Science and Technology Innovation Program for Postgraduate students in IDP subsidized by Fundamental Research Funds for the Central Universities ZY20220302 This work was funded by the National Natural Science Foundation of China under Grant 42007422, and the Science and Technology Innovation Program for Postgraduate students in IDP subsidized by Fundamental Research Funds for the Central Universities under Grant ZY20220302. There was no additional external funding received for this study. The funders had no role in study design, data collection and analysis, decision to publish, or preparation of the manuscript.

==============================
Automatic building extraction from very high-resolution remote sensing images is of great significance in several application domains, such as emergency information analysis and intelligent city construction. In recent years, with the development of deep learning technology, convolutional neural networks (CNNs) have made considerable progress in improving the accuracy of building extraction from remote sensing imagery. However, most existing methods require numerous parameters and large amounts of computing and storage resources. This affects their efficiency and limits their practical application. In this study, to balance the accuracy and amount of computation required for building extraction, a novel efficient lightweight residual network (ELRNet) with an encoder-decoder structure is proposed for building extraction. ELRNet consists of a series of downsampling blocks and lightweight feature extraction modules (LFEMs) for the encoder and an appropriate combination of LFEMs and upsampling blocks for the decoder. The key to the proposed ELRNet is the LFEM which has depthwise-factorised convolution incorporated in its design. In addition, the effective channel attention (ECA) added to LFEM, performs local cross-channel interactions, thereby fully extracting the relevant information between channels. The performance of ELRNet was evaluated on the public WHU Building dataset, achieving 88.24% IoU with 2.92 GFLOPs and 0.23 million parameters. The proposed ELRNet was compared with six state-of-the-art baseline networks (SegNet, U-Net, ENet, EDANet, ESFNet, and ERFNet). The results show that ELRNet offers a better tradeoff between accuracy and efficiency in the automatic extraction of buildings in very highresolution remote sensing images. This code is publicly available on GitHub (https://github.com/GaoAi/ELRNet).

Introduction

With the advancement of remote sensing technology, the automatic extraction of buildings using very high-resolution (VHR) remote sensing images has been widely adopted in city construction (Song et al., 2017; Grinias, Panagiotakis & Tziritas, 2016), earth observation (Wei & Yang, 2020; Moser, Serpico & Benediktsson, 2012), and disaster assessment (Longbotham et al., 2011; Liu et al., 2019a). Accurate and efficient extraction of buildings from the complex background of VHR remote sensing images has become a challenging problem (Mahabir et al., 2018; Li et al., 2019; Lu et al., 2018). In recent years, researchers have used deep learning (DL) methods to study the application of DL algorithms in building extraction from VHR remote sensing images (Yi et al., 2019; Guo et al., 2020; Liu et al., 2019b; Li et al., 2018; Sa et al., 2018; Ran et al., 2021).

For instance, Huang et al. (2016) proposed an end-to-end deep neural network that effectively and robustly extracts buildings of various shapes by fusing multi-source information. Yuan (2017) designed a method to integrate activation features from multiple layers into a fully convolutional network (FCN)-based pixel-level prediction. His approach provides a scalable solution for the automatic extraction of dense houses. Although these methods extract most buildings, the fusion of deep and shallow features is insufficient, which seriously affects accuracy. Therefore, researchers have proposed strategies to enhance building extraction accuracy.

Skip layer connections have been used to fuse deep and shallow features (Badrinarayanan, Kendall & Cipolla, 2017; Sheikh, Maity & Kole, 2022; Chen et al., 2021). Liu et al. (2019a) proposed a novel method, referred to as spatial residual inception network (SRI-Net), which aggregates multi-scale contextual information by successively fusing multi-level features. SRI-Net achieved 89.09% intersection-over-union (IoU) on the Wuhan University (WHU) dataset. Sheikh, Maity & Kole (2022) proposed an infrared U-Net (IRU-Net) to alleviate the semantic gap between encoder and decoder features. This method achieved 81.52% IoU on the Massachusetts building dataset. Chen et al. (2021) proposed a new structure, referred to as dense residual network (DR-Net), which uses a densely connected convolutional neural network and residual network structures to fully integrate the features extracted from the shallow and deep layers of the network. The DR-Net structure achieved 86% IoU on the WHU building dataset. Ji, Wei & Lu, (2018) proposed Siamese U-Net (SiU-Net), based on U-Net (Ronneberger, Fischer & Brox, 2015), which accepts the original image and its downsampled corresponding image as inputs. The two inputs in the network share the same U-Net architecture and the same set of weights which are then concatenated to generate the final result. SiU-Net achieved 88.4% IoU. However, its generalisation ability to multi-source datasets is limited. They then designed a network structure based on multi-scale aggregation and added a radiometric augmentation strategy to extract buildings from different data sources (Ji, Wei & Lu, 2019).

Other researchers optimised the effects of building extraction using post-processing methods (Ma et al., 2020; He et al., 2016). Wei, Ji & Lu (2019) designed a two-stage network which initially extracts buildings from VHR remote sensing images, and then uses regularised polygons to correct the extracted buildings, thereby significantly improving the extraction results. The building residual refine network (BRRNet) proposed in Shao et al. (2020) combines a prediction module and a residual refinement module and achieved an F1-score of 85.36% on the Massachusetts building dataset. Jung, Choi & Kang (2021) proposed an extraction method composed of a holistically nested edge detection module and a boundary enhancement module to elaborate the edges of constructions and optimised the edges in the extraction results.

Although the aforementioned strategies can significantly improve the accuracy of building extraction, they contain numerous parameters, which increases the computational load. For instance, SegNet, BRRNet, and SiU-Net have approximately 29.5, 17, and in excess of 11.04 million parameters, respectively. In fact, the number of parameters of the DL network is a critical indicator in measuring the performance of approaches in practical applications (Jing, Lin & Wang, 2020; Mehta et al., 2018). Hence, many researchers have explored improvement strategies for DL networks to reduce the number of computational parameters and improve efficiency.

The number of network parameters can be reduced by replacing the standard convolution in the residual block with a factorised convolution (FC) (Alvarez & Petersson, 2016). Paszke et al. (2016) used FC to improve the bottleneck residual block and designed a deep neural network referred to as Enet, achieving an IoU of almost 86% with 0.36 million parameters. Romera et al. (2017) used FC for the non-bottleneck residual structure and proposed a lightweight network, referred to as efficient residual factorized network (ERFNet). This network achieved an IoU of 87.24% with 14.67 GFLOPs and 2.06 million parameters. Lo et al. (2019) used an asymmetric dilated convolution structure and dense connectivity to propose efficient dense modules with asymmetric convolution network (EDANet). This method achieved an IoU of 84.75% with 4.44 GFLOPs and 0.68 million parameters. Depthwise separable convolution (Chollet, 2017), which was proposed to reduce the number of parameters during feature extraction, reduces the computational load. A depthwise separable convolution comprises a depthwise convolution (DW) and pointwise convolution (PW). Lin et al. (2019) merged DW and FC into a residual module and added a dilated convolution to design a new efficient model referred to as ESFNet. This method achieved an IoU of 84.98% with 2.55 GFLOPs and 0.18 million parameters. In general, many lightweight architectures increase efficiency at the expense of accuracy (Huang, Chen & Wang, 2021; Yu et al., 2020). The tradeoff between network accuracy and efficiency is a key research point in the design of lightweight building extraction networks; therefore, there is room for further exploration of these networks.

This study aimed to improve the efficiency of building extraction while ensuring high accuracy, thus proposing an efficient lightweight residual network (ELRNet). A sequential encoder–decoder structure with 23 layers was used as the backbone network, and a lightweight feature extraction module (LFEM) was designed as the core feature extraction module, to accurately extract features while reducing the computational load. In particular, the proposed LFEM consists of depthwise-factorised convolution (DFC) and efficient channel attention (ECA). DFC can significantly reduce the computational complexity by factorising the standard convolution and changing the operations of the convolution process. ECA can further improve accuracy by capturing the information of cross-channel interactions with negligible amount of computation. The main contributions of this study are summarised as follows:

(1) A lightweight building extraction network, ELRNet, is proposed in this study. The self-designed lightweight feature extraction module LFEM is embedded in ELRNet, to overcome the disadvantages of error accumulation in the process of downsampling and upsampling, thereby improving the accuracy and efficiency of information interaction between different layers. ELRNet achieves a balance between accuracy and efficiency in automatic building extraction from high-resolution remote sensing images.

(2) The proposed LFEM by adopting DFC to reduce the computational complexity of the feature-extraction process, ensures network efficiency. ECA was added to LFEM, to extract effective information from different channels of high-resolution remote sensing images, which improved the accuracy of the extraction results.

(3) Experimental results show that the proposed ELRNet can achieve competitive accuracy in relation to other state-of-the-art methods on the WHU dataset in several metrics, such as IoU and Params, with better tradeoff between accuracy and efficiency. The proposed ELRNet obtained excellent building extraction results without postprocessing.

The remainder of this article is organised as follows. The Methods section describes the architecture and core components of the proposed ELRNet. The Results section presents a series of comparative experimental analyses and a reasonable discussion. Finally, conclusions are drawn in the last chapter.

Methods

Architecture

This study aimed to construct a novel deep learning network for building extraction, to achieve a better tradeoff between accuracy and efficiency. The proposed efficient lightweight building extraction network, referred to as ELRNet, uses an encoder–decoder structure with 23 layers. A three-band image is fed into two consecutive downsampling blocks to enlarge the feature extraction receptive field and reduce the computational complexity, as shown in Fig. 1. Subsequently, the extracted feature maps are passed through five consecutive LFEMs for further feature extraction. After the feature maps go through a downsampling block and eight LFEMs, feature maps containing rich information are output, which finalises the encoder. To fuse the features between different feature maps and match the resolution of the input image, the extracted feature maps are decoded by passing them through three deconvolution blocks and four LFEMs. Finally, a building extraction image containing pixel-level classification is obtained.

Figure 1 Overall architecture of ELRNet.

The image in the figure is from the WHU dataset, and the data download link is: https://study.rsgis.whu.edu.cn/pages/download/building_dataset.html.

The proposed architecture is detailed in Table 1. The entire framework is that of an encoder–decoder architecture. Taking the input of a three-channel 512 × 512 image as an example, after 16 layers of encoder phase, 128-channel 64 × 64 images are obtained as output feature maps. Following decoder upsampling and fusion stages, these feature maps are finally transformed into a two-channel (i.e., two classes) 512 × 512 image of the same size as the input image. The encoder phase consists of 1–16 layers, and contains three downsampling blocks and 13 LFEMs. Layers 17–23 comprise the decoder, which includes upsampling blocks and LFEMs to further fuse the features between different layers.

Table 1 Details of ELRNet.

The input and output sizes are in the form of C × W × H, where C, W, and H are the number of channels, width, and height of the image, respectively.

	Layer	Blocks	Input size	Output size	
ENCODER	1	Downsampling	3 × 512 × 512	16 × 256 × 256	
	2	Downsampling	16 × 256 × 256	64 × 128 × 128	
	3–7	5 × LFEM	64 × 128 × 128	64 × 128 × 128	
	8	Downsampling	64 × 128 × 128	128 × 64 × 64	
	9–16	8 × LFEM	128 × 64 × 64	128 × 64 × 64	
DECODER	17	Upsampling	128 × 64 × 64	64 × 128 × 128	
	18–19	2 × LFEM	64 × 128 × 128	64 × 128 × 128	
	20	Upsampling	64 × 128 × 128	16 × 256 × 256	
	21–22	2 × LFEM	16 × 256 × 256	16 × 256 × 256	
	23	Upsampling	16 × 256 × 256	2 × 512 × 512	

In the encoder phase, the feature extraction of VHR remote sensing images of the original size requires the calculation of a large number of convolution operations, which incurs a high computational load. The downsampling block can be leveraged in the first few layers of a deep learning network to reduce the number of calculation units. Moreover, considering that downsampling processes reduce the resolution of the feature map, which results in the loss of important details, this study focuses on minimising the use of downsampling. Hence, only the downsampling blocks at layers 1, 2, and 8 are considered, which can alleviate the computational load and ensure the effectiveness of information extraction. In the remaining layers of the encoder, the proposed LFEM is added to perform key feature information extraction with fewer parameters.

The decoder further fuses the features between different layers to match the resolution of the input image. Deconvolution layers with stride two were used for upsampling at layers 17, 20, and 23. Deconvolution does not require sharing of the pooling indexes of the encoder, thus reducing memory and computation requirements. In addition, to enhance feature recovery, LFEM was applied to layers 18–19 and 21–22. The encoder–decoder architecture renders the proposed network lightweight. The downsampling block and proposed LFEM are detailed below.

Downsampling block

The downsampled images have larger receptive fields and more contextual information. For classification tasks, not only the colour and texture of buildings, but also the scenes in which they appear are important. In addition, downsampling can narrow the resolution of the image to reduce computational complexity.

Therefore, a novel downsampling block with two branches was used, as shown in Fig. 2. The branch of max pooling with stride two can compress redundant spatial information in the image into a more effective representation, thereby improving the feature extraction results. In addition, the loss of important feature information caused by premature downsampling was compensated by using ENet (Paszke et al., 2016), whereby a convolution was performed with stride two in parallel with max pooling. Finally, the convolution and pooling results were concatenated to obtain richer feature outputs. Accordingly, it can be deduced that this downsampling block can extract effective feature information and plays an essential role in improving the accuracy and efficiency of the final extraction results.

Figure 2 Downsampling block.

Lightweight feature extraction module (LFEM)

An efficient feature extraction module is crucial for improving network performance. This study focused on designing a lightweight feature extraction module for more effective feature extraction with lower computational complexity. The bottleneck and non-bottleneck structures proposed in Ji, Wei & Lu (2019) are classic feature extraction modules that are widely used in image classification tasks. The two structures are shown in Figs. 3A and 3B. In the third layer of the network, taking the feature map with input and output sizes of 128 and 64 channels as an example, the number of parameters in the bottleneck structure is 53,248, and that of the non-bottleneck structure is 147,456, which is greater by almost a factor of 2.77. The two structures exhibit almost the same precision. However, the bottleneck structure requires fewer computing resources (Romera et al., 2017; Lin et al., 2019). Therefore, the structure of the bottleneck is consistent with the design of the lightweight module and was used as a benchmark for the proposed feature extraction module.

Figure 3 The bottleneck (A) and non-bottleneck (B) residual structures proposed in Ji, Wei & Lu (2019).

The following design process considers the problem of insufficient feature extraction by the bottleneck structures. Therefore, first a bottleneck model with standard convolution (BM-SC) was designed, as shown in Fig. 4A. In the convolution branch, the design of the original bottleneck structure consists of 1 × 1, 3 × 3, and 1 × 1 three-layer convolution operations. The 1 × 1 layers are responsible for reducing and subsequently restoring the dimensions of the feature map. In the residual-mapping branch, a 3 × 3 standard convolution block is added for further feature extraction. BM-SC can compensate for the lack of feature information extraction in lightweight networks. However, the addition of a convolution block increases the computational complexity of BM-SC. To improve the information extraction ability without increasing the computational load, a bottleneck model with depthwise-factorised convolution (DFC), referred to as BM-DFC, was designed to replace the 3 × 3 standard convolution in the convolution and residual mapping branches of BM-SC, as shown in Fig. 4B. DFC is a combination of factorised convolution (Alvarez & Petersson, 2016) and depthwise convolution (Chollet, 2017). In factorised convolution, a standard 2D convolution with 3 × 3 kernels can be converted into two 1D convolutions with kernels of 3 × 1 and 1 × 3. Factorised convolution shrinks the model by reducing the number of redundant parameters and acts as a regulariser in the entire network, enhancing generalisation ability. The key to depthwise convolution is changing the convolution method by setting groups in the convolution process. The operation of DW is illustrated in Fig. 5A. A convolution kernel in DW extracts the feature information of a channel from the input feature map. The number of convolution kernels is similar to the number of channels in the input feature map. In Fig. 5B, compared with the standard convolution operation, DW can clearly reduce the computational complexity. Therefore, DFC is essential for designing a lightweight feature extraction module. Compared with BM-SC, the BM-DFC significantly reduces the number of parameters.

Figure 4 Two residual structures of the design process: (A) BM-SC and (B) BM-DFC feature extraction modules.

Figure 5 (A) Depthwise and (B) standard convolution operations for a three-channel input feature map.

To further improve the accuracy of the final building extraction results, after improving efficiency, ECA (Wang et al., 2020) was used to capture the information of cross-channel interactions with a negligible amount of computation. The architecture of ECA is shown in Fig. 6. The efficient channel attention module first performs a global average pooling operation on the input feature map, which is further subjected to a 1D convolution operation with a convolution kernel size of k to perform information cross-channel interactions. Note that k can be adaptively selected in this process based on the number of channels. The formula for the adaptive selection of k is expressed as follows: (1) t= intabslogc,2+b/γ

(2) k=fC=log2Cγ+bγodd

Figure 6 Flowchart of efficient channel attention—ECA mechanism.

where |t|odd represents the odd number closest to t. Note that γ and b are set to two and one, respectively, as described in Wang et al. (2020).

Equations (1) and (2) indicate that high-dimensional channels are characterised by long-distance interactions, whereas low-dimensional channels are characterised by short-distance interactions. In the following operation, a sigmoid activation function is implemented to obtain the weight of each channel. Finally, the weights are multiplied by the corresponding elements of the original input feature map to obtain the final output feature map. ECA can obtain effective channel information and reduce the impact of noise, which improves feature extraction performance.

Considering the advantages of the ECA module, after the DFC in the convolution branch based on BM-DFC, an ECA model was added. This is because ECA can further refine the feature map, which allows the generation of more effective feature maps for dimension recovery. This new module is referred to as the LFEM lightweight feature extraction module and is the final core module of the proposed network, as shown in Fig. 7. LFEM effectively improves feature extraction capacity with a small number of parameters.

Figure 7 LFEM—lightweight feature extraction module of the proposed design.

Flowchart

The process followed in this study is clarified in the flowchart shown in Fig. 8. In the training stage, the model learns the building features in the remote sensing images based on the training and validation sets, prior to generating the weight model. In the prediction stage, the weight model is applied to the test set for automatic detection of buildings, and relevant evaluation metrics are calculated to measure the quality of the model. When the test results do not attain the expected values, the model is retrained on the datasets until the best prediction results are generated, to obtain the optimal weight model.

Figure 8 The flowchart of this study.

Results

Datasets

The effectiveness of the proposed method was demonstrated on the WHU public building extraction dataset (Ji, Wei & Lu, 2018), comprising data from two cities (New Zealand and Christchurch), with a total area greater than 450 km2. The spatial resolution of the images is 0.3 m. The dataset contains 8,188 optical remote sensing images with a crop size of 512 × 512 pixels and their corresponding label images. The entire dataset was divided into training, validation, and test sets with 4,736, 1,036, and 2,416 images, respectively. All the experiments were conducted using the original partition of this dataset.

Evaluation metrics

To measure the accuracy of the proposed ELRNet for the building extraction tasks, three general metrics were used: IoU, F1-score, and overall accuracy (OA). The IoU calculates the ratio of intersection and union between the predicted and actual segmentation areas. The OA represents the ratio of correctly classified pixels to all pixels. The F1-score is a composite metric of precision and recall. The higher the values of the above three metrics, the better is the model performance. (3) IoU=TPTP+FP+FN

(4) F1−score=2×TP2×TP+FN+FP

(5) OA=TP+TNTP+TN+FP+FN

where TP, FP, FN, and TN are the number of pixels corresponding to true positive, false positive, false negative, and true negative, respectively.

To measure the efficiency of the proposed ELRNet for building extraction, the floating-point number (FLOPs) and number of parameters (Params) were used as evaluation metrics in the network training process. In addition, the time required to complete the predictive inference for each network (TestTime) was evaluated.

Experimental setup

All experiments were implemented using the Pytorch1.11.0 deep learning framework, training and testing the model on workstations with GPU RTX3090. The workstation configuration was based on an Intel(R) Xeon(R) Gold 6230R CPU @ 2.10 GHz, 252 GB memory, 64-bit Ubuntu 20.04 operating system, and 11.3 CUDA.

As optimizer, Adam (Kingma & Ba, 2014) was used with weight decay rate and momentum set to 0.0002 and 0.9, respectively. Validation was performed every 10 epochs and the best-weighted model in the validation set was used as the final model for performance evaluation. The initial parameter details for training and testing are shown in Table 2.

Table 2 Initial parameter details for training and testing.

Parameter name	Initial parameter	
Weight decay rate	0.0002	
Momentum	0.9	
Initial learning rate	0.0005	
Training epochs	300	
Batch size	8	

Ablation studies

A series of ablation experiments was designed to demonstrate the effectiveness of the ELRNet architecture on the WHU dataset and the importance of each main component of the proposed method. The experiments replaced the feature extraction module in the network with the five different modules described in the Lightweight feature extraction module (LFEM) section: non-bottleneck, bottleneck, BM-SC, BM-DFC, and LFEM.

The results of the ablation experiments displayed in Table 3 indicate that the backbone of ELRNet can effectively extract buildings, regardless of the feature extraction module used. ELRNet-BM-SC with almost four times more floating-point calculations compared with ELRNet-bt, increases the accuracy of the extraction results. Compared with ELRNet-bt, the parameter and computational complexities of ELRNet-BM-DFC are reduced by 0.43 and 0.07, respectively, with IoU and F1-scores that are slightly better than those of ELRNet-bt. Furthermore, compared with ELRNet-BM-DFC, the IoU in ELRNet-LFEM increases by 0.6% without a significant increase in computational complexity and number of parameters. This result shows that ECA can improve accuracy without affecting efficiency. Therefore, ELRNet-LFEM achieves a better balance between accuracy and efficiency in building extraction. Examples of visual comparisons of the ablation experiments on the WHU test set are shown in Fig. 9. The proposed ELRNet is insensitive to the shape and density of buildings, and the extraction results are more complete and more consistent with the original label image.

Table 3 Comparative results of ablation studies, where bt and non-bt represent the bottleneck and non-bottleneck structures, respectively.

The best metric value is highlighted in bold.

Networks	FLOPs (GFLOPs)	Params (M)	IoU (%)	F1 (%)	OA (%)	
ELRNet-non-bt	20.97	3.02	89.44	94.43	98.67	
ELRNet-bt	3.34	0.30	87.52	93.34	98.40	
ELRNet-BM-SC	12.72	1.75	88.69	94.01	98.57	
ELRNet-BM-DFC	2.91	0.23	87.64	93.41	98.41	
ELRNet-LFEM	2.92	0.23	88.24	93.75	98.50	

Figure 9 Some comparative examples of ablation experiments on the WHU test dataset.

(A) Original image; (B) label image; (C) ELRNet-non-bt; (D) ELRNe-bt; (E) ELRNet-BM-SC; (F) ELRNet-BM-DFC; (G) ELRNet-LFEM. The images in the figure are from the WHU dataset, and the data download link is https://study.rsgis.whu.edu.cn/pages/download/building_dataset.html.

In summary, the ablation experiments proved that the designed architecture and feature extraction module are effective, and their ingenious combination improves network performance. The proposed ELRNet architecture is advantageous for accurate extraction of buildings with limited hardware resources. In addition, the proposed LFEM feature extraction module can improve the extraction results for buildings in VHR remote sensing images.

Comparative analysis

In this section, the proposed ELRNet is evaluated and compared with two popular baseline networks (SegNet and U-Net) and four state-of-the-art lightweight building-extraction methods (ENet, EDANet, ESFNet, and ERFNet) for the same experimental environment and configuration. The six aforementioned metrics were used to evaluate the effectiveness of the proposed method. The experimental results are displayed in Table 4.

Table 4 Results of the comparison between the proposed method and the state-of-the-art lightweight building extraction networks on the WHU dataset.

The best metric value is highlighted in bold.

Networks	FLOPs (GFLOPs)	Params (M)	IoU (%)	F1 (%)	OA (%)	Test Time (s)	
SegNet (Badrinarayanan, Kendall & Cipolla, 2017)	160.56	29.44	83.03	90.73	97.62	40.34	
U-Net) (Ronneberger, Fischer & Brox, 2015)	124.21	13.40	86.30	92.65	98.13	38.22	
ENet (Paszke et al., 2016)	2.45	0.36	86.48	92.75	98.17	11.16	
EDANet (Lo et al. 2019)	4.44	0.68	84.75	91.74	97.89	11.20	
ESFNet (Lin et al., 2019)	2.55	0.18	84.98	91.88	97.93	10.02	
ERFNet (Romera et al., 2017)	14.67	2.06	87.24	93.18	98.28	12.80	
ELRNet	2.92	0.23	88.24	93.75	98.50	11.13	

The baseline methods achieved the highest accuracy although they require a large number of computations and parameters. By comparing the proposed method with the baseline methods of SegNet and U-Net, it can be deduced that ELRNet reduces the computational complexity by 55 and 43 times and improves the IoU by 5.21% and 1.94%, respectively. In contrast to the efficient ENet architecture, the IoU of the proposed ELRNet increased by 1.76% and the numbers of parameters decreased by 0.13. Moreover, compared with EDANet and ESFNet, the IoU improved by approximately 3.49% and 3.26% , respectively. In addition, compared with ERFNet, the computational complexity of the proposed method is lower by 11.75 GFLOPs, and the IoU of the proposed network achieves an improvement of 1%. The experiments demonstrated that the proposed architecture provides an excellent tradeoff between reliability and speed.

A comparison of the prediction results obtained using the six networks on the WHU test set is shown in Fig. 10. To verify the effectiveness of the proposed ELRNet, images of various types of small, large, special-shaped, dense, and sparse buildings were selected from the test results and used for comparison. It is observed that the proposed network performs well for various types of building extraction, and the boundary extraction of buildings is clearer and more regular. Therefore, the proposed ELRNet can achieve a higher extraction accuracy with lower computational complexity and fewer parameters.

Figure 10 Some comparative examples of the proposed ELRNet and state-of-the-art semantic segmentation networks on the WHU test dataset.

(A) Original image; (B) label image; (C) SegNet; (D) U-Net; (E) ENet; (F) EDANet (G) ERFNet; (H) ESFNet; (I) our ELRNet.

In summary, a novel lightweight network, ELRNet, was designed to balance accuracy and efficiency in the task of automatic building extraction from high-resolution remote sensing images.

The main highlights of ELRNet can be summarised in the following two aspects: (1) Our network is specially designed for the characteristics of VHR images, which have a lower spatial resolution than other data. In view of the low-resolution characteristics, as few layers as possible were selected in the design of the network layer as well as the downsampling layer to ensure that the image would retain its characteristics when reduced in size to the smallest size in the downsampling layer. After experimental exploration, the current 23-layer network design achieved the best test results on the VHR image datasets. (2) VHR images contain a variety of ground objects, and the process of building extraction is easily affected by surrounding similar ground objects. Our self-designed feature extraction module, LFEM, not only reduces the influence of surrounding noise but also emphasises the difference between fine-grained features and location representation of different layers, to extract more effective information.

However, ELRNet also has some limitations. Because the shapes and styles of buildings in remote sensing images in different regions are different, the lightweight network in our study is insufficient to extract detailed feature information of extremely small buildings. In the future, we will continue to explore improvements to the network model, primarily with respect to two aspects: the feature extraction module and spatial attention mechanism.

Conclusions

In this study, an efficient lightweight residual network ELRNet was proposed for building extraction from VHR remote sensing images. The proposed method uses an encoder–decoder structure and leverages the design of the proposed LFEM feature extraction module to achieve a balance between accuracy and efficiency.

The ablation experiments demonstrated that LFEM with depthwise-factorized convolution and an efficient channel attention mechanism can efficiently and accurately extract features. Regardless of the module used as the core feature extraction block, ELRNet-LFEM exhibited a better tradeoff between reliability and speed. The proposed method was evaluated and compared with two popular baseline networks and four state-of-the-art architectures. The experimental results demonstrate that the proposed method can extract various types of buildings with clearer boundaries and a better balance between efficiency and accuracy.

Therefore, the proposed ELRNet lightweight architecture is a better choice for achieving fast and accurate extraction of buildings from high-resolution remote sensing images. Furthermore, the proposed network can be easily promoted and used in practical applications.

Supplemental Information

Supplemental Information 1 Code

Additional Information and Declarations

Competing Interests

Author Contributions

Data Availability

The authors declare there are no competing interests.

Ai Gao conceived and designed the experiments, performed the experiments, analyzed the data, performed the computation work, prepared figures and/or tables, authored or reviewed drafts of the article, and approved the final draft.

Guang Yang conceived and designed the experiments, performed the experiments, analyzed the data, performed the computation work, prepared figures and/or tables, authored or reviewed drafts of the article, and approved the final draft.

The following information was supplied regarding data availability:

The original WHU Building Dataset was available at:

- https://study.rsgis.whu.edu.cn/pages/download/building_dataset.html

- Gao, A. (2023). WHU_Building_dataset [Data set]. Zenodo. https://doi.org/10.5281/zenodo.10043889

- Gao, Ai (2024). WHU building dataset. figshare. Dataset. https://doi.org/10.6084/m9.figshare.25415098.v1

The code is available in the Supplemental File and at GitHub and Zenodo:

- https://github.com/GaoAi/ELRNet

- GaoAi. (2024). GaoAi/ELRNet: ELRNet_2 (ELRNet_2). Zenodo. https://doi.org/10.5281/zenodo.10686494.

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
