# Peer review of "Research on building extraction from remote sensing imagery using efficient lightweight residual network"

_PeerJ Computer Science, doi:10.7717/peerj-cs.2006_

## Round 0.1 · original submission · Major Revisions

Dear authors,

Reviewers have now commented on your work. Your article has not been recommended for publication in its current form. We encourage you to clearly address the concerns and criticisms raised by reviewers and resubmit your revised article once you have updated it accordingly.

Best wishes,

**Language Note:** The review process has identified that the English language must be improved. PeerJ can provide language editing services - please contact us at copyediting@peerj.com for pricing (be sure to provide your manuscript number and title). Alternatively, you should make your own arrangements to improve the language quality and provide details in your response letter. – PeerJ Staff

Reviewer 1 ·

Basic reporting

No comments

Experimental design

No comments

Validity of the findings

No comments

Additional comments

This article conducts research on the contradiction between the accuracy and
computational efficiency of existing building remote sensing image extraction, proposes
a lightweight network structure, and provides a clear explanation of the network
structure. The article has clear logic, easy-to-understand English expression, exquisite
graphic representation, and concise and clear table content, which meets the
requirements of the journal. Identified the source of the data and provided the original
code. The experimental design is reasonable, and the results are presented, which are
compared with existing advanced network structures. Through practice, it has been
proven that this network structure can effectively balance extraction accuracy and
computational efficiency.


But there are still some issues that need further improvement:
(1) The dataset and evaluation strategy should be placed under the method title;
(2) Add discussion content, clearly propose small conclusions, and analyze the strengths
and weaknesses of this work
(3)Regarding the list of references: The format should be unified, such as 39,40. Some
references are too old, for example 4, 5. Is there any new alternative literature?

Annotated reviews are not available for download in order to protect the identity of reviewers who chose to remain anonymous.

Reviewer 2 ·

Basic reporting

In the study, researchers conducted research on building extraction from remote sensing images using a lightweight residual network. In this direction, a novel network has been proposed and coding-encoding structures are included in this network architecture. Researchers have clearly stated the problem and developed a new network as a solution. The study contains enough novelty. However, eliminating the some issues will increase the readability and quality of the study.

- In the last part (paragraph) of the Introduction, what is explained in other sections should be briefly mentioned. In short, the organization of the article should be shown.
- Researchers should also pay attention to spelling rules. There are spelling errors in some places. For example, commas are not used in numerical expressions. For instance in line 219, 8188 should be written as 8,188, etc.

Experimental design

- The researchers have written the methods neatly and understandably. However, the flow chart of the study is not provided. The flow chart of the study should be given and the scheme should be briefly summarized.
- The researchers used IoU, accuracy and F1-score values to evaluate the model. What is IoU? Why was this metric used? What is the reason for using this metric? Similarly, why were accuracy and F1-score metrics evaluated? These should be explained under the relevant section.

Validity of the findings

- Various results are given in Table 3 but not interpreted. What do these results mean? The results must be interpreted and conveyed to the reader. What do IoU, OA and F1-score show us? What should a good IoU score be? These must be interpreted. In cases where the data set is unbalanced, the accuracy score alone is not a sufficient evaluation criterion. Generally, specificity, sensitivity and F1-score are used. For reasons like these, these evaluation methods used should be interpreted.
- The advantages and disadvantages of the study should be given in items under the section of Comparative and Analysis. What are the highlights of this model? Which problem in the literature was solved by this study? They should be discussed. Similarly, possible limits of the model should also be mentioned and possible problems should be discussed.

---

## Round 0.2 · accepted · Accept

Dear authors,

Thank you for the revision and for clearly addressing all the reviewers' comments. I confirm that the paper is improved and addresses the concerns of the reviewers. Your paper is now acceptable for publication in light of the last revision.

Best wishes,

Reviewer 1 ·

Basic reporting

no comment

Experimental design

no comment

Validity of the findings

no comment